# Active Learning in Bayesian Neural Networks: Balanced Entropy Learning Principle

## Abstract

Acquiring labeled data is challenging in many machine learning applications with limited budgets. Active learning gives a procedure to select the most informative data points and improve data efficiency by reducing the cost of labeling. The info-max learning principle maximizing mutual information such as BALD has been successful and widely adapted in various active learning applications. However, this pool-based specific objective inherently introduces a redundant selection. In this paper, we design and propose a new uncertainty measure, Balanced Entropy Acquisition (BalEntAcq), which captures the information balance between the uncertainty of underlying softmax probability and the label variable. To do this, we approximate each marginal distribution by Beta distribution. Beta approximation enables us to formulate BalEntAcq as a ratio between a shifted entropy and the marginalized joint entropy. The closed-form expression of BalEntAcq facilitates parallelization by estimating two parameters in each marginal Beta distribution. BalEntAcq is a purely standalone measure without requiring any relational computations with other data points. Nevertheless, BalEntAcq captures a well-diversified selection near the decision boundary with a margin, unlike other existing uncertainty measures such as BALD, Entropy, or Mean Standard Deviation (MeanSD). Finally, we demonstrate that our balanced entropy learning principle with BalEntAcq consistently outperforms well-known linearly scalable active learning methods, including a recently proposed PowerBALD, a simple but diversified version of BALD, by showing experimental results obtained from MNIST, CIFAR-100, SVHN, and TinyImageNet datasets.

## 1 Introduction

Acquiring labeled data is challenging in many machine learning applications with limited budgets. As the dataset size gets bigger and bigger for training a complex model, labeling data by humans becomes more expensive. Active learning gives a procedure to select the most informative data points and improve data efficiency by reducing the cost of labeling.

The active learning problem is well-aligned with a subset selection problem that can find the most efficient but minimal subset from the data pool [70, 34, 18, 66, 86, 87, 85]. The difference is that active learning is typically an iterative process where a model is trained and a collection of data points is selected to be labeled from an unlabelled data pool. Therefore, it is still a theoretically very challenging but important problem.

It is now commonly accepted that standard deep learning models do not capture model uncertainty correctly. The simple predictive probabilities are usually erroneously described as model confidence [31]. So there is a risk that a model can be misdirecting its outputs with high confidence. However, the predictive distribution generated from Bayesian deep learning models better captures the uncertainty

from the data [26, 51, 67, 17]. Therefore, we focus on developing an active learning framework in the Bayesian deep neural network model by leveraging the Monte-Carlo (MC) dropout method as a proxy of the Gaussian process [26] which may facilitate further analysis.

## 1.1 Our contributions

Our proposed active learning method is well-aligned with Bayesian experimental design [89, 14, 80, 62, 24] with an assumption that the forward active learning iterative process follows the Bayesian prior-posterior framework. Furthermore, our approach is also aligned with Bayesian uncertainty quantification methods [40, 1, 35, 41, 26, 27, 48, 67, 47] with an assumption that the working neural network model is a Bayesian network [49].

In this paper, we extend and improve recent advances in both aspects of Bayesian experimental design and Bayesian uncertainty quantification. We investigate the generalized notion of the joint entropy between model parameters and the predictive outputs by leveraging a point process entropy [64, 25, 73, 16, 5]. By approximating the marginals using Beta distributions, we then derive an explicit formula of the marginalized joint entropy by estimating Beta parameters from Bayesian deep learning models. As a Bayesian experiment, we revisit the well-known entropy and mutual information measures given expected cross-entropy loss. We show that well-known acquisition measures are functions of marginal distributions through analytical formulas. We propose our new uncertainty measure, Balanced Entropy Acquisition (BalEntAcq), which captures the information balance between the uncertainty of underlying softmax probability and the label variable. Finally, we demonstrate that our balanced entropy learning principle with BalEntAcq consistently outperforms well-known linearly scalable active learning methods, including a recently proposed PowerBALD [47] for mitigating the redundant selection in BALD [27], by showing experimental results obtained from MNIST, CIFAR-100, SVHN, and TinyImageNet datasets.

## 2 Background

### 2.1 Problem formulation

We write an unlabeled dataset $\mathcal{D}_{\textbf{pool}}$ and the labeled training set $\mathcal{D}_{\textbf{training}} \subseteq \mathcal{D}_{\textbf{pool}}$ in each active learning iteration. We denote by $\mathcal{D}_{\textbf{training}}^{(n)}$ if it's necessary to indicate the specific $n$-th iteration step. Given $\mathcal{D}_{\textbf{training}}$, we train a Bayesian deep neural network model $\Phi$ with model parameters $\omega \sim \mathbf{p}(\omega)$.

Then for a data point $\mathbf{x}$ given $\mathcal{D}_{\textbf{training}}$, the Bayesian deep neural network $\Phi$ produces the prediction probability: $\Phi(\mathbf{x}, \omega) := (P_1(\mathbf{x}, \omega), \cdots, P_C(\mathbf{x}, \omega)) \in \Delta^C$ where $\Delta^C = \{(p_1, \cdots, p_C) : p_1 + \cdots + p_C = 1, p_i \geq 0 \text{ for each } i\}$ and $C$ is the number of classes. For the final class output $Y$, it is assumed to be a multinoulli distribution (or categorical distribution):

$$Y(\mathbf{x}, \omega) := \begin{cases} 1 & \text{with probability } P_1(\mathbf{x}, \omega) \\ \vdots & \vdots \\ C & \text{with probability } P_C(\mathbf{x}, \omega). \end{cases} \tag{1}$$

For the sake of brevity, we sometimes omit $\mathbf{x}$ or $\omega$ by writing $\Phi(\omega)$, $P_i(\omega)$, $Y(\omega)$ or $\Phi$, $P_i$, $Y$ unless we need further clarifications on each data point $\mathbf{x}$. Under this formulation, the oracle (active learning algorithm) selects a subset of data points to add to the next training set, i.e. at $(n+1)$-th iteration, the training set is determined by $\mathcal{D}_{\textbf{training}}^{(n+1)} = \mathcal{D}_{\textbf{training}}^{(n)} \cup \{\text{Next training batch from Oracle}\}$. Once the next training batch is selected, the selected batch will be labeled. This means that the ground truth label information of the selected data is added in training set $\mathcal{D}_{\textbf{training}}^{(n+1)}$ in the next round. Then the goal in active learning is to minimize the number of selected data points to reach a certain level of prediction accuracy.

### 2.2 Examples of uncertainty based active learning methods

In this section, we list up well-known uncertainty measures suitable for Bayesian active learning.

1. **Random**: $\text{Rand}[\mathbf{x}] := U(\omega')$ where $U(\cdot)$ is a uniform distribution which is independent to $\omega$. Random acquisition function assigns a random uniform value on $[0, 1]$ to each data point.

2. **BALD** (Bayesian active learning by disagreement) [58, 35, 27]: $\text{BALD}[\mathbf{x}] := \Im\left(\omega, Y\left(\mathbf{x}, \omega\right)\right)$, where $\Im(\cdot, \cdot)$ represents a mutual information between random measures. BALD captures the mutual information between the model parameters and the predictive output of the data point. In practice, we calculate the mutual information between the predictive output and the predictive probabilities.

3. **Entropy** [82]: $\text{Ent}[\mathbf{x}] := -\sum_i \left(\mathbb{E}P_i\right) \log\left(\mathbb{E}P_i\right)$. Entropy is the Shannon entropy with respect to the expected predictive probability. Entropy can be the uncertainty of the prediction probability. Moreover, under the cross-entropy loss, we may also interpret the entropy measure as an expected loss gain since $-\log\left(\mathbb{E}P_i\right)$ is the cross-entropy loss given the ground truth label is the class $i$.

4. **Mean standard deviation (MeanSD)** [14, 40, 1]: $\text{MeanSD}[\mathbf{x}] := \frac{1}{C}\sum_i \sqrt{\mathbb{E}P_i^2 - \left(\mathbb{E}P_i\right)^2}$. Mean standard deviation captures the average of the standard deviations for each marginal distribution.

5. **PowerBALD** [21, 47]: $\text{PowerBALD}[\mathbf{x}] := \log \text{BALD}[\mathbf{x}] + Z$, where $Z$ is an independently generated random value from Pareto distribution with the exponent $\alpha > 0$. We use $\alpha = 1$ as a default choice suggested by [47]. The motivation of this randomized acquisition is to mitigate the redundant selection by diversifying selected multi-batch points. In general, we do not know which exponent will be the optimal choice.

In a multiple acquisition scenario, we simply add the above uncertainty values for each data point $\mathbf{x}_i$:

$$\text{AcqFunc}[\mathbf{x}_1, \cdots, \mathbf{x}_n] := \sum_{i=1}^{n} \text{AcqFunc}[\mathbf{x}_i], \tag{2}$$

where $\text{AcqFunc} \in \{\text{Rand}, \text{BALD}, \text{Ent}, \text{MeanSD}, \text{PowerBALD}\}$.

## 2.3 Summary of other active learning approaches

Cohn et al. [14] provided one of the first statistical analyses in active learning, establishing how to synthesize queries that reduce the model's forward-looking error by minimizing its variance leveraging MacKay's closed-form variance approximation [60]. In this fashion, there exists a line of works in Bayesian experimental design [11, 58, 89, 14, 80, 90, 24, 23, 38] with an assumption that the forward active learning iterative process follows Bayesian prior-posterior framework.

On the other hand, in active learning, accommodating both the information uncertainty and the diversification of the acquired samples is essential to improve the performance under multi-batch acquisition scenarios. In a theoretical perspective, the most natural way to combine the uncertainty and the diversification seems to leverage reasonable sub-modular functions, e.g. Nearest neighbor set function [92], BatchBALD [48], Determinantal Point Process [6] and SIMILAR [50] with sub-modular information measures, and then/or apply a fast linear-time algorithm to find a diversified multi-batch with a provable performance guarantee [69, 70, 20, 95, 79, 36, 37, 57]. Although a fast linear-time solver is available for general sub-modular functions, there still exists a gap with practical implementation, such as high memory requirements, which makes the computation unscalable for identifying multi-batch acquisition points, e.g., BatchBALD [48]. Similar to the sub-modular function optimization, there exist many customized optimization approaches, e.g. CoreSet [81] and more approaches [29, 39, 19, 94, 91].

Another recent approach is to look at parameters of the neural network and to diversify points such as BADGE [4] with gradients and BAIT [3] with Fisher information. There also exist network architectural design focused approaches such as Learning loss by designing loss prediction layers [96], UncertainGCN and CoreGCN [8] with graph neural networks , VAAL [84] and TA-VAAL [42] by applying adversarial learning methods.

# 3 Bayesian neural network model

We adopt the Bayesian neural network framework introduced in Gal et al. [26]. The core idea in the Bayesian neural network is leveraging the MC dropout feature to generate a distribution of the predictive probability as an output at inference time. Under mild assumptions, it turns out that it is equivalent to an approximation to a Gaussian Process [77, 68, 93, 26, 56].

## 3.1 Each softmax probability marginal approximately follows Beta distribution

We may consider a Bayesian neural network model $\Phi$ as a random measure, i.e., stochastic process parametrized by $\mathcal{D}_{\mathbf{training}}$ over the data set $\mathcal{D}_{\mathbf{pool}}$. Given a data point $\mathbf{x} \in \mathcal{D}_{\mathbf{pool}}$, $\Phi(\mathbf{x}, \omega)$ produces a random probability distribution in a simplex $\Delta^C$. This analogy has a close connection with the construction of random discrete distribution, originally introduced by Kingman [45]. Since then, random measure construction has been extensively developed in Bayesian nonparametrics, and it is well-known that Dirichlet probability having Beta marginals plays the central role in the construction of the random discrete distribution [46, 22, 75, 74, 7, 72, 78]. It is the main motivation of the Beta distribution approximation. Many kinds of literature similarly assume the Dirichlet distribution after the softmax in the Bayesian neural network.

As illustrated by Milios et al. [65], we may follow the construction of Dirichlet distribution. Following the approach by Ferguson [22], a Dirichlet probability can be constructed through a collection of independent Gamma distributions. On the other hand, each marginal in Gaussian Process (approximated by Bayesian neural network) in the softmax output having dependent components follows a log-normal distribution (before the normalization, but after the exponentiation in softmax). Then by applying the shape similarity between a log-normal distribution and Gamma distribution, the construction of random probability from log-normal distributions would produce an approximated Dirichlet distribution. Therefore we may assume that the marginal distribution would approximately follow the Beta distribution.

Alternatively, as an analytical approach, we may see that Beta approximation can be justified through Laplace approximation [61, 32, 33, 17]. There exists a mapping between multivariate Gaussian distribution and Dirichlet distribution under a softmax basis. Then Beta distribution follows as a marginal distribution of Dirichlet distribution. Therefore we may assume that Beta approximation exists through Laplace approximation under the assumption that the Bayesian neural network produces the multivariate Gaussian distribution (as a marginalized Gaussian process over finite rank covariate function) before the softmax layer [68, 93, 26, 56].

In practice, once we estimate the sample mean and sample variance for each marginal of $\Phi(\mathbf{x}, \omega)$, we can estimate two parameters of the Beta distribution as follows. Assume that $P_i \sim \mathrm{Beta}\,(\alpha_i, \beta_i)$. If $\mathbb{E}P_i = m_i$ and $\mathrm{Var}P_i = \sigma_i^2$, then

$$\alpha = \frac{m^2(1-m)}{\sigma^2} - m, \quad \beta = \left(\frac{1}{m} - 1\right)\alpha. \tag{3}$$

When $P_i \sim \mathrm{Beta}\,(\alpha_i, \beta_i)$, $\mathbb{E}P_i = \frac{\alpha_i}{\alpha_i + \beta_i} = m$ and $\mathrm{Var}P_i = \frac{\alpha_i \beta_i}{(\alpha_i + \beta_i)^2(\alpha_i + \beta_i + 1)} = \sigma_i^2$. Solving the equation with respect to $\alpha_i$ and $\beta_i$, then the (3) follows.

## 3.2 Marginalized joint entropy in Bayesian neural network

In this section, we derive a marginalized joint entropy in the Bayesian neural network, which shall be further discussed in constructing our main results. We may formulate the Bayesian neural network $\Phi$ as a well-known encoder-decoder framework. The sender sends a message $(\mathbf{x}, \omega)$ with a random key $\omega$ through the Bayesian neural network, then the receiver receives a message $Y(\mathbf{x}, \omega)$.

Under this framework, controlling $\omega$ is difficult, but we can control the family of the encoded messages $\Phi(\mathbf{x}, \omega)$ in a tractable manner [27, 43, 88]. We can easily prove that the mutual information between $\omega$ and $Y$ is the same as the mutual information between the encoded $\Phi(\mathbf{x}, \omega)$ and the predictive output $Y$ since $Y$ depends only on $\Phi(\mathbf{x}, \omega)$:

$$\mathrm{BALD}[\mathbf{x}] := \mathfrak{I}\,(\omega, Y(\mathbf{x}, \omega)) = H(Y(\mathbf{x}, \omega)) - \mathbb{E}_\omega\,[H\,(Y(\mathbf{x}, \omega)\,|\omega)] \tag{4}$$

$$= H(Y(\mathbf{x}, \omega)) - \mathbb{E}_\Phi\,[H\,(Y(\mathbf{x}, \omega)\,|\Phi(\mathbf{x}, \omega))] = \mathfrak{I}\,(\Phi(\mathbf{x}, \omega), Y(\mathbf{x}, \omega)), \tag{5}$$

where $H(Y(\mathbf{x}, \omega))$ represents the Shannon entropy by marginalizing out the randomness of $\omega$ in $Y(\mathbf{x}, \omega)$ and $\mathfrak{I}(\cdot, \cdot)$ represents a mutual information between two quantities.

The formulations of the mutual information (4) - (5) look natural, but we need to note that $\omega$ or $\Phi(\mathbf{x}, \omega)$ is on a continuous domain, and $Y(\mathbf{x}, \omega)$ is on a discrete domain. This combined domain implies that we cannot directly apply Shannon entropy and differential entropy notions [15]. One immediate question is what the joint entropy between $\Phi(\mathbf{x}, \omega)$ and $Y(\mathbf{x}, \omega)$ is. For this, we can leverage point process entropy [64, 25, 73, 16, 5] by generalizing the notion of the entropy

in this combined domain. We consider the joint entropy of $\Phi\left(\mathbf{x}, \omega\right)$ and $Y(\mathbf{x}, \omega)$, denoting by $\mathfrak{H}\left(\Phi\left(\mathbf{x}, \omega\right), Y(\mathbf{x}, \omega)\right)$ through the point process entropy. We write a Janossy density function [16] $j\left(\mathbf{p}, y = i\right)$ of $\left(\Phi\left(\mathbf{x}, \omega\right), Y(\mathbf{x}, \omega)\right)$ on $\Delta^C \times [C]$ as follows:

$$j\left(\mathbf{p}, y = i\right) = p_i f\left(\mathbf{p}\right), \tag{6}$$

where $\mathbf{p} := \left(p_1, \cdots, p_C\right)$ and $f(\cdot)$ is a density function of $\Phi\left(\mathbf{x}, \omega\right)$. Then the joint entropy of $\Phi\left(\mathbf{x}, \omega\right)$ and $Y(\mathbf{x}, \omega)$ can be defined as

$$\mathfrak{H}\left(\Phi\left(\mathbf{x}, \omega\right), Y(\mathbf{x}, \omega)\right) = -\sum_{i=1}^{C} \int_{\Delta^c} j\left(\mathbf{p}, y = i\right) \log j\left(\mathbf{p}, y = i\right) \mathrm{d}\mathbf{p}. \tag{7}$$

By plugging (6) into (7), we have the following identity.

$$\mathfrak{H}\left(\Phi\left(\mathbf{x}, \omega\right), Y(\mathbf{x}, \omega)\right) = H(Y\left(\mathbf{x}, \omega\right)) + \mathbb{E}_Y\left[h\left(\Phi\left(\mathbf{x}, \omega\right) | Y\left(\mathbf{x}, \omega\right)\right)\right], \tag{8}$$

where $H(\cdot)$ represents the usual Shannon entropy, and $h(\cdot)$ represents the usual differential entropy. By applying Jensen's inequality, we may derive a marginalized joint entropy as an upper bound of the joint entropy:

$$\mathfrak{H}\left(\Phi\left(\mathbf{x}, \omega\right), Y(\mathbf{x}, \omega)\right) \leq -\sum_i \mathbb{E}_{P_i}\left[P_i \log\left(P_i f(P_i)\right)\right], \tag{9}$$

where we ambiguously write $f(\cdot)$ to be a density function for each $P_i$. Assume that each $P_i \sim \text{Beta}(\alpha_i, \beta_i)$ by applying Beta approximation. We then define a quantity of the marginalized joint entropy from (9) and we find an equivalent formulation as follows:

$$\text{MJEnt}[\mathbf{x}] := -\sum_i \mathbb{E}_{P_i}\left[P_i \log\left(P_i f(P_i)\right)\right] = \underbrace{\sum_i \left(\mathbb{E}P_i\right) h(P_i^+)}_{\text{posterior uncertainty}} + \underbrace{H(Y)}_{\text{entropy}}, \tag{10}$$

where $P_i^+$ is the conjugate Beta posterior entropy of $P_i$ which follows $P_i^+ \sim \text{Beta}(\alpha_i + 1, \beta_i)$. We remark that $h(P_i^+)$ can be easily calculated by the closed form entropy formula of Beta distribution. i.e.

$$h(P_i^+) = \log B(\alpha_i + 1, \beta_i) - \alpha_i \Psi(\alpha_i + 1) - (\beta_i - 1)\Psi(\beta_i) - (\alpha_i + \beta_i - 1)\Psi(\alpha_i + \beta_i + 1),$$

where $B(\cdot, \cdot)$ is the Beta function, and $\Psi(\cdot)$ is the Digamma function. We call the first term in (10) to be the posterior uncertainty. We may interpret the posterior uncertainty as an expected posterior entropy assuming that we observed a positive sample of the class toward $P_i$ for each $i$ without knowing the true class label. The first term is always non-positive, and is maximized (equals to 0) when each $P_i^+$ is $\text{Beta}(1, 1)$, i.e., Uniform on $[0, 1]$. So $-\infty < \text{MJEnt}[\mathbf{x}] \leq H(Y)$. The second entropy term can be decomposed into two uncertainty terms:

$$H(Y) = \underbrace{\mathfrak{I}\left(\omega, Y\right)}_{\text{epistemic uncertainty}} + \underbrace{\mathbb{E}_\omega\left[H\left(Y | \omega\right)\right]}_{\text{aleatoric uncertainty}}. \tag{11}$$

The epistemic uncertainty captures the model uncertainty (as BALD), and the aleatoric uncertainty captures the data uncertainty [63]. Therefore the marginalized joint entropy, MJEnt[$\mathbf{x}$] is a decomposition of three types of uncertainty values.

### 3.3 Entropy is for maximizing an expected cross-entropy loss

Given a ground-truth label $\{Y = i\}$, the cross-entropy loss of the neural network can be given as $\text{loss}\left(\Phi\left(\mathbf{x}, \omega\right), Y = i\right) = -\log \mathbb{E}P_i$. Therefore we can calculate the expected cross-entropy loss without knowing the truth label:

$$\text{ExpectedLoss}[\mathbf{x}] := \sum_{i=1}^{C} \mathbb{P}\left[Y = i\right] \text{loss}\left(\Phi\left(\mathbf{x}, \omega\right), Y = i\right) = -\sum_i \left(\mathbb{E}P_i\right) \log\left(\mathbb{E}P_i\right) = \text{Ent}[\mathbf{x}].$$

Based on the re-formulation, we may interpret that entropy acquisition is for maximizing an expected cross-entropy loss in a selection of acquisition points, aligning the idea with the learning loss [96]. The natural question is, "Once we acquire a data point that maximizes entropy acquisition, can we remove/or learn this expected cross-entropy amount of loss at the future stage of the active learning?". The answer could be "No." The exhaustive loss acquisition could only happen if the neural network perfectly over-fits the training data. Therefore, there exists a gap between a realistic neural network training scenario and the objective of the entropy acquisition. Our equivalent loss interpretation gives us an insight into why the entropy acquisition might not be successful in practice, even in the single-point acquisition scenario.

### 3.4 BALD is a function of marginals and is strongly aligned with maximizing an expected cross-entropy loss difference upto the next iteration

We have the mutual information between $\omega$ and $Y$ and it is the same as the mutual information between the encoded message and the channel output since $Y$ depends only on $\Phi(\mathbf{x}, \omega)$ [27]:

$$\text{BALD}[\mathbf{x}] := \mathfrak{I}(\omega, Y) = \mathfrak{I}(\Phi(\mathbf{x}, \omega), Y(\mathbf{x}, \omega)), \tag{12}$$

where $\mathfrak{I}(\cdot, \cdot)$ represents mutual information between two quantities. By assuming that $\Phi(\mathbf{x}, \omega)$ follows a Dirichlet distribution, we can calculate the mutual information analytically [2]. Then by investigating further into the analytical mutual information formula, we see that the marginal distributions $P_i$'s in $\Phi(\mathbf{x}, \omega)$ are sufficient to estimate BALD. Therefore we can state BALD through Beta marginal distributions as follows. See Appendix for more details.

**Theorem 3.1.** *Under Beta marginal distribution approximation, let $P_i \sim Beta(\alpha_i, \beta_i)$ in $\Phi(\mathbf{x}, \omega)$. Then the mutual information BALD$[\mathbf{x}]$ can be estimated as follows:*

$$\begin{aligned}
BetaMarginalBALD[\mathbf{x}] := &\sum_{i=1}^{C} (\alpha_i - 1)\Psi(\alpha_i + \beta_i) - \sum_{i=1}^{C} \left(\frac{\alpha_i}{\alpha_i + \beta_i}\right)\log\left(\frac{\alpha_i}{\alpha_i + \beta_i}\right) - \sum_{i=1}^{C} \frac{\alpha_i(\alpha_i - 1)}{\alpha_i + \beta_i}\Psi(\alpha_i) \\
&- \sum_{i=1}^{C} \frac{\beta_i(\alpha_i - 1)}{\alpha_i + \beta_i}\Psi(\alpha_i + \beta_i + 1) + \sum_{i=1}^{C} \left(\frac{\alpha_i^2}{\alpha_i + \beta_i}\right)[\Psi(\alpha_i + 1) - \Psi(\alpha_i + \beta_i + 1)].
\end{aligned}$$

As a Bayesian experimental design process, we may assume that each Beta marginal distribution $P_i$ with the ground-truth label $\{Y = i\}$ of the next trained model would follow the Beta posterior distribution $P_i^+$. Without this assumption, existing choices of acquisition functions such as BALD or MeanSD might not be well-justified. For example, what is the implication of maximizing mutual information through the active learning process with a Bayesian neural network? How is it different from the maximization of the entropy acquisition? To answer these questions, leveraging our Beta marginalization and considering the similar idea of expected information gain [24], we may consider the expected cross-entropy loss difference between the current stage model and the next stage model.

$$\begin{aligned}
\text{ExpectedEffectiveLoss}[\mathbf{x}] := &\sum_{i=1}^{C} \mathbb{E}P_i \left[-\log \mathbb{E}P_i - \left(-\log \mathbb{E}P_i^+\right)\right] \\
= &\sum_{i=1}^{C} \left(\frac{\alpha_i}{\alpha_i + \beta_i}\right)\left[\log\left(\frac{\alpha_i + 1}{\alpha_i + \beta_i + 1}\right) - \log\left(\frac{\alpha_i}{\alpha_i + \beta_i}\right)\right].
\end{aligned}$$

ExpectedEffectiveLoss captures the effective amount of cross-entropy loss for the model to learn after the acquisition. By definition, we see that ExpectedEffectiveLoss aims to exclude the undesirable over-fitting scenario assumption unlike Entropy acquisition.

Since Digamma function $\Psi(x) \sim \log x - \frac{1}{2x}$ where $f(x) \sim g(x)$ implies $\lim_{x \to \infty} f(x)/g(x) = 1$, we may expect that BetaMarginalBALD$[\mathbf{x}]$ and ExpectedEffectiveLoss$[\mathbf{x}]$ would behave similarly. Figure 1 shows the Spearman's rank correlations among different acquisition measures upto a class dimension $C = 10,000$. We observe that BetaMarginalBALD behaves equally like the original BALD and we confirm that BALD and MeanSD are strongly aligned with maximizing ExpectedEffectiveLoss. Therefore, acquiring points through BALD or MeanSD could be a better strategy than Entropy because BALD or MeanSD takes into account the effective loss acquisition instead of the unrealistic full amount of the loss acquisition.

## 4 Balanced entropy learning principle

The previous section shows that well-known acquisition measures have an objective toward the cross-entropy loss, and they are closely related to marginal distributions. According to Farquhar et al. [21], to be successful in active learning, they hypothesize that it is crucial to find a good balance between active learning bias and over-fitting bias under over-parametrized neural networks. In parallel to their hypothesis, we define the balanced entropy (BalEnt) to be a ratio between the marginalized joint entropy (9) and the shifted entropy:

$$\text{BalEnt}[\mathbf{x}] := \frac{\text{MJEnt}[\mathbf{x}]}{\text{Ent}[\mathbf{x}] + \log 2} = \frac{\sum_i (\mathbb{E}P_i)h(P_i^+) + H(Y)}{H(Y) + \log 2}. \tag{13}$$

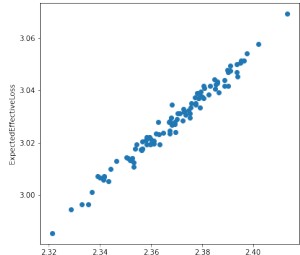 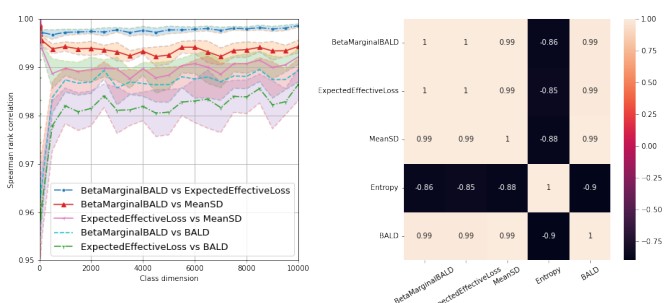

Figure 1: Scatter plot at $C = 10,000$ between BALD and ExpectedEffectiveLoss (left), Spearman's rank correlations over various class dimensions (middle), and Spearman's rank correlation matrix at $C = 10,000$ (right). The relationship between BetaMarginalBALD and ExpectedEffectiveLoss consistently captures a high rank-correlation with $> 99.6\%$ regardless of the class dimensions. BALD and ExpectedEffectiveLoss show $> 97.5\%$ rank-correlation. We randomly generate $100$ softmax applied $C$-dimensional Gaussian samples and repeated the process $10$ times. Shaded band shows the standard deviation.

Recall that we call the first term in MJEnt[$\mathbf{x}$] to be posterior uncertainty, and it is an expected posterior entropy of underlying marginals. BalEnt captures the information balance between the posterior uncertainty from the model $\Phi$ and entropy of the label variable $Y$.

## 4.1 Implications of balanced entropy

To understand the implication of BalEnt[$\mathbf{x}$], we can prove the following Theorem 4.1.

**Theorem 4.1.** *Let* $\Delta^{-1} := \lfloor 2e^{H(Y)} \rfloor$ *and* $\Upsilon := \{I_n\}$, *a collection of evenly divided intervals in* $[0, 1]$ *where* $I_n := \big[(n-1)\Delta, n\Delta\big)$ *for* $n = 1, \cdots, (\Delta^{-1} - 1)$ *and* $I_{\Delta^{-1}} := [1 - \Delta, 1]$. *Let* $\bar{P}_i$ *be a discretized random variable over* $\Upsilon$ *of* $P_i$ *from* $\Phi(\mathbf{x}, \omega)$. *For any estimator* $\hat{P}_i$ *of* $\bar{P}_i$ *given the label* $\{Y = i\}$ *we have*

$$\mathbb{E}\left[ \mathbb{P}\left[ \hat{P}_i \neq \bar{P}_i \middle| Y = i \right] \right] \geq \frac{\sum_i (\mathbb{E}P_i)\, h(P_i^+) + H(Y)}{H(Y) + \log 2}(1 + \epsilon_1) - \epsilon_2 = BalEnt[\mathbf{x}](1 + \epsilon_1) - \epsilon_2,$$

*where* $\epsilon_1, \epsilon_2 \geq 0$ *are adjustment terms depending on* $\Delta$ *such that* $\epsilon_1 \to 0$ *and* $\epsilon_2 \to 0$ *as* $\Delta \to 0$.

Theorem 4.1 tries to answer the following inverse problem. For the unlabeled data point, $\mathbf{x}$, if we know the information of the label $\{Y = i\}$, how much can we reliably estimate the underlying probability $P_i$ from the model $\Phi$? As we know that $-\log P_i$ is the cross-entropy loss of the trained model with $Y$, it equivalently answers the estimation error probability of the loss prediction under a unit precision up to $-\log \Delta$ level. For the precision level, we are assuming to carry $-\log \Delta \approx H(Y) + \log 2$ nats - natural unit of information, re-scaled amount of bits, matching the enumerator with MJEnt[$\mathbf{x}$] term. It is not clear how to determine a better choice of the precision level $-\log \Delta$. But we may understand the denominator $H(Y) + \log 2$ is for normalizing the term BalEnt[$\mathbf{x}$] $\leq 1$ as a probability. Then the sign of BalEnt[$\mathbf{x}$] becomes very important. BalEnt[$\mathbf{x}$] $\geq 0$ implies that it could be impossible to perfectly predict the loss $-\log P_i$ given currently available information. i.e., there could exist information imbalance between the model and the label approximately starting from BalEnt[$\mathbf{x}$] $= 0$. Therefore, insight from Theorem 4.1 suggests us a new direction for our main active learning principle. We define our primary acquisition function, namely, balanced entropy learning acquisition (BalEntAcq), as follows:

$$\text{BalEntAcq}[\mathbf{x}] := \begin{cases} \text{BalEnt}[\mathbf{x}]^{-1} & \text{if BalEnt}[\mathbf{x}] \geq 0, \\ \text{BalEnt}[\mathbf{x}] & \text{if BalEnt}[\mathbf{x}] < 0, \end{cases}$$

Since the information imbalance exists at least from BalEnt[$\mathbf{x}$] $= 0$, we prioritize to fill the information gap from BalEnt[$\mathbf{x}$] $= 0$ toward positively increasing direction. If we try to fill the information imbalance gap from the highest BalEnt[$\mathbf{x}$], the information imbalance would still exist around BalEnt[$\mathbf{x}$] $= 0$ area. Therefore, it might not improve the active learning performance much. See Appendix A12.2 and A12.3 for different prioritization and precision level results. That's the motivation why we take the reciprocal of BalEnt[$\mathbf{x}$] when BalEnt[$\mathbf{x}$] $\geq 0$.

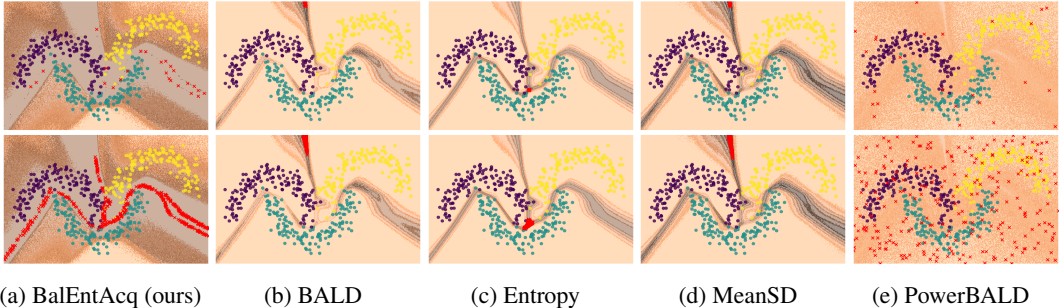

| (a) BalEntAcq (ours) | (b) BALD | (c) Entropy | (d) MeanSD | (e) PowerBALD |

Figure 2: Top-$K$ selected points are marked by red color. The first row shows the top $K = 25$ points. The second row shows the top $K = 500$ point selections among around 0.6 million grid points.

## 4.2 Toy example illustration

To illustrate the behavior of BalEntAcq and its relationship with other uncertainty measures, we train a simple Bayesian neural network with a 3-class moon dataset in $\mathbb{R}^2$. Then we calculate each acquisition measure for all fixed lattice points in the square domain by assuming that the unlabeled pool is highly regularized (or uniform). i.e., by evenly discretizing the domain, we obtain each uncertainty value for each lattice point. The total number of lattice points is around 0.6 million. Then we choose top-$K$ high uncertainty values for each method to observe the prioritized region for each method. We use $K = 25$ and $K = 500$. Figure 2 illustrates the top-$K$ points selected by each method. The most significant phenomenon is that BalEntAcq's selection is highly diversified near the decision boundary showing a bifurcated margin because we are prioritizing the surface area of $\{\text{BalEnt}[\mathbf{x}] \geq 0\}$. This is well-aligned with the strategy avoiding high aleatoric points. (See Appendix A.13) Then we can imagine to conduct a uniform sampling on each contour surface $\{\text{BalEnt}[\mathbf{x}] = \lambda\}$ for each $\lambda \geq 0$, as we move to the surface for each $\lambda < 0$. That's why we observe bifurcated but diversified and balanced selection near the decision boundary with BalEngAcq in Figure 2-(a) when $K = 25$. On the other hand, there is a preferred area for each method from other measures except PowerBALD. PowerBALD shows a good diversification, but it could select non-informative points.

## 5 Experimental Results

In this section, we demonstrate the performance of BalEntAcq from MNIST [55], CIFAR-100 [52], SVHN [71], and TinyImageNet [54] datasets under various scenarios. We used a single NVIDIA A100 GPU for each experiment, and details about the experiments are explained in Appendix A.12. We test Random, BALD, Entropy, MeanSD, PowerBALD, and BalEntAcq measures. We add BADGE for additional baseline. Note that all acquisition measures except BADGE in our experiments are standalone quantities, so all can be easily parallelized.

**Single acquisition active learning with MNIST.** MNIST [55] is the most popular and elementary dataset to validate the performance of image-based deep learning models initially. We use a simple convolutional neural network (CNN) model applying dropouts to all layers with a single acquisition size. The primary purpose of this single acquisition experiment is to validate our proposed balanced entropy approach by removing the contribution of diversification unlike multi-batch acquisition scenario.

**Fixed features with CIFAR-100 and 3×CIFAR-100.** In recent years, significant efforts have been made on building an efficient framework of unsupervised or self-supervised feature learning such as SimCLR [12, 13], MoCo [30], BYOL [28], SwAV [9], DINO [10], etc. As an application in active learning, we may leverage the feature space from the unsupervised feature learning without explicitly knowing true labels but construct a good representation space. In our experiments, we adopt SimCLR [12] for simplicity with ResNet-50 to build a feature space for CIFAR-100.

With 3×CIFAR-100 dataset, we observe the effect of the redundant information treatment for each method by adding three identical points. We use the same fixed feature obtained from SimCLR with CIFAR-100. We may observe how each method effectively diversifies the selection under a redundant data pool scenario by fixing the feature space.

**Pre-trained backbone with SVHN and strong data augmentation with TinyImageNet.** In this experiment, we follow a typical image classification scenario in practice. We use the ResNet-18 backbone for SVHN and the ResNet-50 backbone for TinyImageNet with ImageNet pre-trained model for model architecture, and the last linear classification layer is replaced with a simple Bayesian neural network with dropouts. We apply strong data augmentations for TinyImageNet, including random crop, random flip, random color jitter, and random grayscale. Under this scenario, the feature space from the backbone is continuously evolving and keeps confused as the training and active learning process proceeds. Because of the strong data augmentation and batch normalization in ResNet-18 or ResNet-50, the decision boundary keeps confused, implying that the Bayesian experimental design assumption might not hold. However, we still want to observe the general behavior of each measure and how to improve the accuracy under a more dynamic feature space.

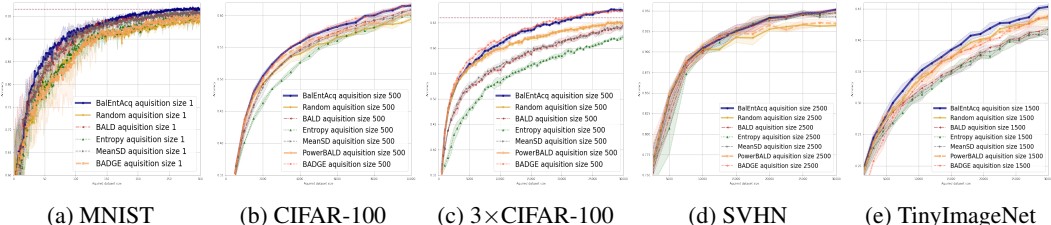

| (a) MNIST | (b) CIFAR-100 | (c) 3×CIFAR-100 | (d) SVHN | (e) TinyImageNet |

Figure 3: Active learning accuracy curves obtained from various scenarios. Our proposed BalEntAcq outperforms well-known acquisition measures, and we repeated the experiment 3 times.

| Scenario | Full dropouts + CNN | | | Fixed feature | | Redundant images + Fixed feature | | Backbone | | Backbone + Augmentation | |
|---|---|---|---|---|---|---|---|---|---|---|---|
| Dataset/Acq. Size/Test size | MNIST/1/10,000 | | | CIFAR-100/500/10,000 | | 3×CIFAR-100/500/10,000 | | SVHN/2,500/26,032 | | TinyImageNet/1,500/10,000 | |
| Train Size/Pool Size | 50/60,000 | 100/60,000 | 300/60,000 | 5,000/50,000 | 10,000/50,000 | 15,000/150,000 | 30,000/150,000 | 15,000/73,257 | 30,000/73,257 | 15,000/100,000 | 30,000/100,000 |
| Random | 78.6 ± 4.9% | 86.4 ± 2.7% | 93.6 ± 0.7% | 55.5 ± 0.4% | 59.4 ± 0.5% | 61.9 ± 0.2% | 64.9 ± 0.3% | 91.8 ± 0.6% | 93.2 ± 0.2% | 37.1 ± 0.3% | 43.8 ± 0.1% |
| BALD | 82.6 ± 1.3% | 90.5 ± 0.8% | 95.3 ± 0.4% | 56.2 ± 0.5% | 60.8 ± 0.3% | 58.8 ± 0.2% | 64.6 ± 0.6% | 92.5 ± 0.8% | 94.8 ± 0.2% | 35.2 ± 0.7% | 41.8 ± 0.4% |
| Entropy | 77.4 ± 2.6% | 87.7 ± 2.0% | 94.8 ± 0.3% | 54.9 ± 0.4% | 60.0 ± 0.3% | 56.7 ± 0.8% | 62.3 ± 0.4% | **92.6 ± 0.4%** | 94.8 ± 0.2% | 35.1 ± 0.4% | 41.8 ± 0.4% |
| MeanSD | 83.4 ± 2.2% | 90.6 ± 1.1% | 96.0 ± 0.2% | 56.0 ± 0.1% | 60.9 ± 0.4% | 59.4 ± 0.5% | 64.3 ± 0.3% | 92.5 ± 0.6% | 94.3 ± 0.2% | 34.7 ± 0.4% | 40.9 ± 0.6% |
| PowerBALD | - | - | - | 56.5 ± 0.1% | 60.3 ± 0.2% | 62.2 ± 0.2% | 65.0 ± 0.7% | 92.2 ± 0.6% | 93.5 ± 0.2% | 37.4 ± 0.7% | 43.4 ± 0.3% |
| BADGE (not-scalable) | 77.0 ± 6.1% | 86.5 ± 4.2% | 94.8 ± 0.4% | **57.4 ± 0.1%** | **61.8 ± 0.1%** | **64.0 ± 0.2%** | **67.4 ± 0.1%** | **92.9 ± 0.4%** | 95.0 ± 0.3% | 37.2 ± 0.6% | 43.9 ± 0.3% |
| BalEntAcq (ours) | **85.4 ± 1.0%** | **91.4 ± 1.3%** | **96.5 ± 0.1%** | 57.2 ± 0.2% | 61.5 ± 0.2% | 63.5 ± 0.5% | **67.4 ± 0.1%** | 92.5 ± 0.8% | **95.2 ± 0.1%** | **38.5 ± 0.2%** | **45.3 ± 0.4%** |

Table 1: Selected accuracy table. Mean and standard deviation are from 3 repeated experiments.

**Discussion.** BalEntAcq consistently outperforms other linearly scalable baselines in all datasets, as shown in Table 1. BADGE performs similarly with Entropy under a single acquisition scenario in MNIST because BADGE focuses on maximizing the loss gradient similar to Entropy, as we explained in Section 3.3. BADGE shows better performances at first when we fix the feature space, but our BalEntAcq eventually merges with the performance of BADGE. We also note that BADGE is not a linearly scalable method. Under dynamic feature scenarios in SVHN or TinyImageNet, we observe that our BalEntAcq performs better. Considering the acquisition calculation time (see Appendix A.14), our BalEntAcq should be a better choice. Figure 3 shows the full active learning curves. For CIFAR-100 and 3×CIFAR-100 cases, by fixing features, we control/remove all other effects possibly affecting the model's performance, such as data augmentation or the role of backbone in the classification. As demonstrated in Figure 2, BalEntAcq is very efficient in selecting diversified points along the decision boundary. Instead, PowerBALD suffers from improving accuracy because it focuses more on diversification/randomization by missing the information near the decision boundary. For SVHN or TinyImageNet, BalEntAcq shows better performance again. We suppose that diversification near the decision boundary in BalEntAcq also plays the data exploration because the representation space keeps evolving with the backbone training.

# 6 Conclusion

In this paper, we designed and proposed a new uncertainty measure, Balanced Entropy Acquisition (BalEntAcq), which captures the information balance between the underlying probability and the label variable through Beta approximation with a Bayesian neural network. BalEntAcq offers a diversified selection and is unique compared to other uncertainty measures. Moreover, we expect that our proposed balanced entropy measure does not have to be confined to active learning problems in general. BalEntAcq can be applied to improve the diversified selection process or accuracy estimation in a different type of Bayesian neural network frameworks. Therefore, we look forward to having further follow-up studies with broad applications beyond the active learning problems.

## Limitations

As we specified in the introduction, our focus is MC-dropout-based Bayesian neural networks; our experiments have been limited to dropout-based Bayesian neural networks. However, our theoretical development does not require special architectural assumptions if we can apply Beta approximation. So one can apply our proposed method to any Bayesian classification network with Beta approximation. Moreover, considering the similarity with recent theoretically guaranteed active learning algorithm with abstention [59, 83, 76, 97] (see Appendix A.13), we expect to replicate the similar out-performance in other types of the Bayesian networks, e.g., Gaussian process [77], ensemble network [53], variational-dropout network [44], Laplace Redux [17], and so on.

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
