# OpenReview forum: "Active Learning in Bayesian Neural Networks: Balanced Entropy Learning Principle"
_NeurIPS.cc/2022/Conference — NeurIPS 2022 Submitted_

### Official Review · Reviewer_xmoA · 2022-07-11

**Rating:** 4
**Confidence:** 3
**Soundness:** 2 fair
**Presentation:** 3 good
**Contribution:** 2 fair

**Summary:**

This paper proposes a new acquisition function BalEntAcq for Al in BNN. It captures the information balance between the uncertainty of underlying softmax probability and the label variable. It facilitates parallelization over all classes (by estimating distribution parameters in marginal distribution). It is a purely standalone measure that could reduce relational computation like BatchBALD, CoreSet, and BADGE. Nevertheless, the paper shows that BalEntAcq captures a well-diversified selection near the decision boundary. The authors also demonstrate the new principle can outperforms other well-known Bayesian active learning algorithms.

**Questions:**

1) Discuss computational complexity and compare it with other AL algorithms.
2) Can the authors explain why we can not apply non-Baysian AL algorithms to BNNs?

**Limitations:**

See 3 weaknesses.

**Strengths And Weaknesses:**

Strengths:
1) The new acquisition function is novel. It facilitates the parallelization both over classes and batch selection (there is no relational computation within data points inside a batch). It captures both informativeness (uncertainty) and diversity.

Weaknesses:
1) Baselines are limited. (a) The paper mentioned BatchBALD. The computational cost for selecting a batch is proportional to the batch size. It is still tolerable. As shown in the paper, it is superior to BALD in many settings. This paper can compare its method with BatchBALD with a small batch size, e.g., batch size <= 40. (2) Variation-Ratio [1] is also a potential baseline that have shown competitive performance. (3) Non-Bayesian AL algorithms can be applied to BNN. For example, in order to apply CoreSet to BNN, we can take the average last-layer representations with N MC dropouts or simply one representation with MC dropout, which is also the case for any DNN with dropout layers. Similarly, we can also apply BADGE to BNN. For all the examples, we can still maintain the expressiveness of the BNNs.
2) The authors only tried their method on MC dropout. There are many other BNNs, e.g., [2] and [3], that need to be considered since the paper focus on AL in BNNs.
3) Datasets are limited. The datasets considered in the paper should cover different aspects of AL, e.g., batch size, data format, and dataset size. The authors can compare their method with BatchBALD and [4] on MNIST and Repeated-MNIST with small batch size, i.e., <= 10. The author can also consider some small UCI datasets and some larger batch sizes for datasets like CIFAR10 and SVHN. If the new method does not work well in these settings, the authors should discuss it in the limitation or conclusion section.

References:
[1] Kirsch, Andreas, Joost Van Amersfoort, and Yarin Gal. "Batchbald: Efficient and diverse batch acquisition for deep Bayesian active learning." Advances in neural information processing systems 32 (2019).
[2] Lakshminarayanan, Balaji, Alexander Pritzel, and Charles Blundell. "Simple and scalable predictive uncertainty estimation using deep ensembles." Advances in neural information processing systems 30 (2017).
[3] Kingma, Durk P., Tim Salimans, and Max Welling. "Variational dropout and the local reparameterization trick." Advances in neural information processing systems 28 (2015).
[4] Kirsch, Andreas, Tom Rainforth, and Yarin Gal. "Test Distribution-Aware Active Learning: A Principled Approach Against Distribution Shift and Outliers." arXiv preprint arXiv:2106.11719 (2021).

---

> ### Author Response · Authors · 2022-08-02
> **Thank you. There's a huge misunderstanding about our intention to set baselines.**
>
> Thank you for reviewing our paper. We see that the reviewer *xmoA* didn't validate any of our theoretical development. **We request to focus on our theoretical contribution.**
>
> First of all, there's a huge misunderstanding about our intention to set baselines.
> - We didn't make any statement like "we can not apply non-Baysian AL algorithms to BNNs" in our paper. **It's the reviewer's wrong perception.** The second question is not a valid request.
> - We have chosen BALD, Entropy, MeanSD, and PowerBALD because they are linearly scalable standalone measures. So it's a **FAIR** comparison.
> - As a response, we added BADGE in our main experiments to better understand our BalEntAcq performance. We also added SVHN in our main experiments. Please check our updated paper in Section 5.
>    - In Appendix A.15, we added $3\times$MNIST and $3\times$CIFAR-10 datasets with **a smaller acquisition size**. In these experiments, we added Variational Ratios, BADGE, BatchBALD, and CoreSet. **We don't see any performance drops of our BalEntAcq in other acquisition sizes.**
>
> Second, we have tested various datasets with MNIST, CIFAR-100, $3\times$CIFAR-100, and TinyImageNet. **We cannot understand why they are limited choices of datasets.**
> - Regarding the weakness of data formats, **it's not a valid point, again.** Please check Appendix A.12.5 to get an idea of the input data format.
>   - For example, our fixed feature experiments such as CIFAR-100 or $3\times$CIFAR-100 are equivalent to tabular datasets.
>       - ResNet-50 with SimCLR [1] generates 2048 dimensional tensor from each image. So the entire training dataset is $50000\times 2048$ sized tabular data.
>   - MNIST is a single-channel gray image dataset. $1\times28\times 28$ size.
>   - TinyImageNet is a three-channel realistic RGB image dataset. $3\times64\times 64$ size.
>   - With the above datasets, we tested the performance of our BalEntAcq with different dropout-based BNN architectures.
>
> Third, regarding the experiments with other Bayesian neural networks, **we clearly stated that we would focus on Dropout-based BNNs in Introduction.**
>   - L37-39. "we focus on developing an active learning framework in the Bayesian deep neural network model by leveraging the Monte-Carlo (MC) dropout method as a proxy of the Gaussian process [26], which may facilitate further analysis."
>   - Nevertheless, in response, we stated in the Limitation section that our experiments had been limited to dropout-based BNNs.
>     - But, our theoretical development does not require special architectural assumptions if we can apply Beta approximation. So one can apply our proposed method to any Bayesian classification network with Beta approximation.
>
> Lastly, we added a time complexity analysis in Appendix A.14.
>   - For BADGE, we tested it with k-means++ instead of k-DPP. In either case, BADGE requires a more expensive computational cost.
>   - Instead, our proposed method BalEntAcq is a linearly-scalable solution. Performance is comparable with BADGE. **In SVHN, TinyImageNet, our BalEntAcq performs better.**
>
> [1] Chen, Ting, et al. "A simple framework for contrastive learning of visual representations." International conference on machine learning. PMLR, 2020.

---

> > ### Comment · Reviewer_xmoA · 2022-08-04
> > **Thank you for your response**
> >
> > 1) Thank you for adding the non-Bayesian AL model, BADGE, to the experiments and explaining why you do not compare with it in the first place. I agree that BADGE is not ``linearly scalable standalone measure'' and computational complexity is high. CoreSet and BADGE are strong baselines. It is better to include them in the experiment to show the performance gap even though they have high computational complexity. Recent AL algorithms, e.g., Cluster-Margin (large batch size), SIMILAR (imbalance and rare classes), and TypiClust (low budget regime), consider a diverse set of datasets.
> > 2) The authors added another dataset SVHN to the main experiment.
> > 3,4) The authors added discussion and complexity analysis.
> >
> > Some concerns about the baselines are solved. The rating is raised accordingly.

---

> > > ### Author Response · Authors · 2022-08-09
> > > **Thank you very much for your follow-up.**
> > >
> > > We included more experiments in the revision. Please check both the main draft and supplementary materials. Here's the summary.
> > > * **Single acquisition BADGE in MNIST** - See Figure 3 and Table 1 in the main article
> > >    - We added a single acquisition BADGE result in MNIST. **We observe that the performance of BADGE is very similar to the entropy. So it's worse than our BalEntAcq.** This is reasonable because BADGE focuses on maximizing the loss gradient similar to Entropy, as explained in Section 3.3.
> > >  * **Another BNN experiment with Variational dropouts** - See Appendix A.18
> > >     - We added Bayesian neural network experimental results with $3\times$CIFAR-10 when variational dropouts [1] have been applied in Appendix A.18. Again, we observe a consistent outperformance compared to other linearly-scalable methods. Also, our BalEntAcq eventually merges with the performance of BADGE.
> > >     - This additional BNN result should be very important because it implies our method's extendability to any other BNN frameworks.
> > >     - Therefore, we hope this addition mitigates your concerns about the extendability to other BNNs.
> > >
> > > * **Comparison with other baselines** - no addition
> > >     - Thank you for suggesting to add more baselines. *We see that this is an additional request after the initial review.* Of course, it's helpful to include many other baselines. However, **given the limited time and computation resources, comparing our method with many other existing methods is beyond our scope.** We do not have a computational infrastructure to train a large batch with a large image dataset of active learning like cluster-margin. We recognize that TypiClust is a recent work, but focusing on a better initial selection is beyond our scope. From our point of view, it's NOT a fair comparison with other selection methods like Figure 4 [2], i.e., we should match the same initial selections for all methods. That being said, we can share our performance expectations of SIMILAR.
> > >     - SIMILAR is a generalized framework with submodular functions. But **figuring out the best combination of the embedding and the target submodular function is still open to the user.** Naive choice of submodular function should not perform better.
> > >        - For example, if we focus on the feature embedding and the Euclidean distance similarity with *LogDet/MI*, the performance should be similar to CoreSet, as we already presented in Appendix A.15. LogDet/MI is equivalent to the selection of k-DPP (determinantal point process) method. Still, it only focuses on the diversification of feature space with Euclidean distance.  As the authors of [3] suggested, it is important to accommodate both the decision boundary and the diversification with the k-DPP method. So the performance improvement in active learning with feature embedding and the euclidean distance similarity should be limited because it does not consider any information about the decision boundary (or loss function).
> > >        - If we focus on the submodular joint mutual information, it should be the same as BatchBALD as we presented in Appendix A.15.
> > >        - If we focus on the gradient embedding, it would be similar to BADGE. So we don't see any good reason to compare it further.
> > >        - As Figure 6 of the SIMILAR paper [4] suggested, the performance of LogDet is worse than BADGE in CIFAR-10.
> > >
> > > We hope that this helps you mitigate your further concerns. Thanks again for taking the time for our paper.
> > >
> > > [1] Kingma, Durk P., Tim Salimans, and Max Welling. "Variational dropout and the local reparameterization trick." Advances in neural information processing systems 28 (2015).
> > >
> > > [2] Hacohen, Guy, Avihu Dekel, and Daphna Weinshall. "Active learning on a budget: Opposite strategies suit high and low budgets.", ICML 2022.
> > >
> > > [3] Bıyık, Erdem, et al. "Batch active learning using determinantal point processes." arXiv preprint arXiv:1906.07975 (2019).
> > >
> > > [4] Kothawade, Suraj, et al. "Similar: Submodular information measures based active learning in realistic scenarios." Advances in Neural Information Processing Systems 34 (2021): 18685-18697.

---

### Official Review · Reviewer_fhr7 · 2022-07-12

**Rating:** 3
**Confidence:** 4
**Soundness:** 2 fair
**Presentation:** 2 fair
**Contribution:** 2 fair

**Summary:**

This paper proposes a new uncertainty measure, called BalEntAcq (Balanced Entropy Acquisition), that aims to correct the potential "bias" of existing active learning schemes based on information-theoretic uncertainty measures, such as the popular BALD (Bayesian active learning by disagreement).
The paper shows that the proposed BalEntAcq captures the "information balance" between the uncertainty of the underlying softmax probability and the label, thereby outperforming other active learning schemes based on similar entropy-based principles.
Based on popular benchmarks, the authors demonstrate the potential advantages of active learning based on BalEntAcq compared to other alternatives.



**Questions:**

1. How does the performance of BalEntAcq compare to the popular ELR strategy, which directly focuses on reducing model uncertainty via active learning so as to improve the predictive performance?

2. How does the proposed active learning scheme perform, in comparison with other existing schemes, in a small-data regime, where the role of active learning would be most important?

3. Does BalEntAcq provide any convergence guarantee - either theoretically or empirically? What would be its main advantages over recently proposed Bayesian active learning schemes?


**Limitations:**

The authors do not explicitly discuss the limitations of the proposed work. The authors note there are no specific "negative societal impacts" relevant to the proposed method.


**Strengths And Weaknesses:**

STRENGTHS

1. The proposed uncertainty measure "BalEntAcq", which aims to effectively capture information balance between the uncertainty of the underlying softmax probability and the label variable, has been shown to result in well-diversified data point selection near the decision boundary, consistently outperforming other information-theoretic uncertainty measures often utilized for active learning.

2. The closed-form expression of the proposed BalEntAcq leads to a computationally efficient active learning scheme, which nevertheless may improve the learning outcomes compared to other existing alternatives.


WEAKNESSES

1. The overall presentation - both literary and technical - requires improvement. For example, the authors do not clearly explain nor define what type of "bias" this study is focusing on, although this is crucial for understanding the main motivation of the proposed scheme as well as the major contributions in this work. Furthermore, many acronyms should be clearly defined before using them.

2. Literature review on existing relevant schemes is too short and shallow. For example, although the authors present a brief summary of existing active learning schemes in Sec. 2.3, the treatment and review of other schemes is not comprehensive and fairly sketchy to be informative for the readers.

3. While BalEntAcq has been shown to result in improved performance over some alternatives, a more comprehensive evaluation is needed to clearly demonstrate the advantage of the proposed scheme. For example, how does it compare to various active learning schemes based on ELR (expected loss reduction) strategy, which focuses on maximally reducing the expected error, and thereby focus on acquiring data points that can maximize the uncertainty directly impacting the predictive performance?

4. Current results show that the proposed BalEntAcq may have potential benefits over other alternatives in terms of efficiency (through closed-form expression of the acquisition function) and performance gain (as reflected in Figure 3 and Table 1). However, the uncertainty measure BalEntAcq is defined in a somewhat heuristic manner, and it is unclear whether it would provide meaningful advantages over recently proposed Bayesian active learning schemes - in terms of performance, robustness, and convergence properties.

For example, the following papers propose Bayesian active learning schemes based on ELR strategies, which have shown to effectively reduce model uncertainty that directly affect the predictive performance of the learned model.

Zhao et al. "Uncertainty-aware Active Learning for Optimal Bayesian Classifier", ICLR 2021.

Tan et al. "Diversity Enhanced Active Learning with Strictly Proper Scoring Rules", NeurIPS 2021.

Both papers provide theoretical convergence guarantees to the optimal model as well as consistent performance improvements compared to other existing schemes, including BALD, the main method to which BalEntAcq is compared in this study. It would be helpful if the authors could compare their scheme to the above methods or other recent Bayesian active learning schemes and discuss what would be the main benefits of the proposed scheme.

5. Based on Figure 3, it is unclear whether BalEntAcq would provide meaningful advantages over other alternatives in the small-data regime, which would be of special interest when studying active learning schemes.

---

> ### Author Response · Authors · 2022-07-31
> **Thank you. We have serious concerns about suggested papers.**
>
> Thank you for reviewing our paper. Here's our overview of the contributions of the suggested two papers.
>
> **We conclude that [1] showed contrived convergence results.** So insights from [1] and its descendent [2] are entirely misleading. Mathematically, Theorem 1 in [1] has no contribution at all. It assumes the finite size of the data and finite parameter space. So there's no room to play with infinite limits. Under the finite setting, we can always guarantee convergence because at some finite time $T$, **any active learning algorithm will acquire all data points.** This implies that any algorithm can achieve the optimal classifier in a finite time.
>
> Moreover, even if we follow the proof, **we cannot guarantee uniform convergence under a countably (or uncountably) infinite data space $X$**, which is the key component of the proof of Theorem 1 in [1]. Therefore all the notions of "almost sure" and "infinitely often" are misleading. Under the finite domain, those infinitely often sets are measure-zero and empty sets. So logically, Lemma 5 in [1] is just *vacuously true*.
>
> [2] also relies on the finite size domain assumption, so all proofs were built on top of the finite space borrowing the idea from [1]. Therefore, the convergence proof in [2] does not mean anything. Unlike the claim in [2], there's no surprise that authors' notion of BALD in [2] converges.
>
> Consequently, **we have serious concerns about the contributions of those papers [1] and [2].**
>
> Here're more comments:
> - Small-data regime: What's the point of active learning in the small-data regime? If the dataset size is small (let's say <1000), the cost of active learning is typically higher than the cost of getting all label information. And if the model has sufficient expressiveness, it will eventually acquire the entire cross-entropy loss. So applying entropy acquisition should be enough under a single acquisition scenario.
>
> - In [2], the authors misinterpreted the definition of BALD under the Bregman divergence framework. BALD is the mutual information between the model and the output for *a single data point*, NOT the weighted average of the mutual information over the entire domain. Therefore the explained BALD in [2] is neither the BALD nor BatchBALD (joint mutual information).
>
> - Nevertheless, since the acquisition idea is still valid, we tested the CoreMSE method [2], seemingly the best under this category, with MNIST under a single acquisition scenario. It showed a better performance than BALD but worse than our BalEntAcq. See **Appendix A.17** in our updated supplementary material.
>     - With a large dataset size, the computational costs of ELR, CoreLog, and CoreMSE are very expensive. They require a vast memory size unless we apply size reductions on the data space $X'$ and MC samples as the authors of [2] did - https://github.com/davidtw999/BEMPS. When the number of classes is large, it is impossible to run the code. Therefore **the naive application of the ELR-based algorithm is not scalable.**
>
> As a final remark, we clarified the term "bias" to "redundant" in our revision. Thank you very much for this suggestion.
>
> [1] Zhao, Guang, et al. "Uncertainty-aware active learning for optimal Bayesian classifier." International Conference on Learning Representations (ICLR 2021). 2021.
>
> [2] Tan, Wei, Lan Du, and Wray Buntine. "Diversity Enhanced Active Learning with Strictly Proper Scoring Rules." Advances in Neural Information Processing Systems 34 (2021): 10906-10918.

---

> > ### Comment · Reviewer_fhr7 · 2022-08-09
> > **Response to rebuttal**
> >
> > Thank you for the response to the review comments.
> >
> > 1. Regarding the comparison against ELR schemes, while I agree with certain points argued by the authors, it would be beneficial to include at least a brief discussion between the proposed BalEntAcq with other acquisition functions used in the ELR strategy.
> > If the authors wish to do so, they may choose a few representative ELR schemes of their choice, and discuss the pros/cons/limitations of these schemes in comparison with BalEntAcq.
> >
> > 2. The "small-data" setting occurs frequently in various scientific domains, including life sciences and materials science, where the labeling cost might be formidable due to the time/cost/labor that would be needed for performing experiments to label data points.
> > It appears that the authors mainly have "computational cost" in mind - which may be why they mention that "the cost of active learning is typically higher than the cost of getting all label information" - but this is not necessarily true in various science domains.
> >
> > 3. It appears that the authors unfortunately misunderstood the "convergence" results in the two papers mentioned.
> > The main issue is whether the acquisition of the labels for the data points selected based on a given acquisition function will make the model learned thereupon converge to the "true" model.
> > In the case of classification, this "model" will be an (unknown) feature-label distribution, which needs to be inferred from the data.
> > Given the posterior of the parameters defining a feature-label distribution, the optimal classifier for this uncertain feature-label distribution will be the OBC (optimal Bayes classifier).
> > And the mentioned papers show that their classifier converges to this OBC.
> >
> > If the authors believe that the convergence guarantees of these recent Bayesian active learning schemes are meaningful *only* in restrictive settings, it would be beneficial to include a brief discussion in the manuscript on this aspect for future readers.
> > Also provide insights into the convergence of the model learned via the data points labeled using the proposed acquisition function.

---

> > > ### Author Response · Authors · 2022-08-09
> > > **Thank you again for the follow-up.**
> > >
> > > **We are afraid to say that both convergence proofs have severe logical flaws, which are unacceptable for us in any case.** Again, the convergence claim *on the null set* is vacuously true. *The reviewer *fhr7* must show a counter-example that does NOT converge to the true model in a finite-data regime.* Otherwise, there's nothing to prove.
> > >
> > > We reject all of the follow-up requests. It's *our courtesy* to include the CoreMSE result in our Appendix as a response to the request in the rebuttal. The convergence proof is nothing to do with our main results. **We firmly emphasize that it's not our standard to include inconsistent results in our main article.**

---

### Official Review · Reviewer_ugGu · 2022-07-12

**Rating:** 3
**Confidence:** 3
**Soundness:** 2 fair
**Presentation:** 2 fair
**Contribution:** 2 fair

**Summary:**

The context is Bayesian active learning, using MC dropout-based models. The paper places Beta distributions over the class marginals (ie one for each $p(y=i|x, w)$). The paper states that BALD as it is usually computed cannot be computed as such and instead proposes a different formulation, which it upper-bounds using what the paper calls a *marginalized joint entropy (MJEnt)* based on the entropy of the Beta distribution approximations. It then introduces a balanced entropy as the ratio between the MJEnt and the regular prediction entropy. Finally, it uses either this value or the reciprocal as the acquisition score (depending on the sign). The paper tries to motivate all this as principled and reports encouraging experimental results.


**Questions:**

1. How does BalEntAcq compare to BABA?

2. Appendix A.12.2 contains ablations. Could you provide an ablation for P4 = BalEnt[x]^-1.

It would seem that BalEntAcq's second case (for negative scores) does not matter as long as there are sufficient positive scorers. This raises the question: it is needed at all even for the experiments provided in the main paper?

**Limitations:**

--

**Strengths And Weaknesses:**

The paper presents a few interesting perspectives on BALD. The experiments show good performance on CIFAR-100 and superior performance on CIFAR-100x3 and TinyImageNet.

I was aware of the papers "BABA: Beta Approximation for Bayesian Active Learning" and "Analytic Mutual Information in Bayesian Neural
Networks" (Anonymous, 2021 &2022), but not the authors. Comparing the text of these papers, there might seem to be a case of self-plagiarism. While the former only exists as a preprint, the latter is to be published in ISIT 2022 (as also noted in this paper). I'm flagging this for the AC to decide and take a look at. The portions in question are not restricted to the Background section of this paper.

In regards to the content of the paper:

1. It is not clear why the Beta approximation makes sense. Indeed, only the marginals $p(y=i | x)$ could be motivated to follow a Beta distribution, similar to the central limit theorem, as these are marginalized over $w$. The $p(y=i|x, w)$ as r.v. of $w$ however, need not follow such a distribution. This is a well-known criticism of e.g. prior networks (Malinin et al, 2018), which fit Dirichlet distributions.

2. The paper states in (4) and (5) that the common formulation of BALD is equal to the more complex formulation using $\Phi$. It then finds an upper bound to this formulation (MJEnt). This is similar to how the predictive entropy (Ent) upper-bounds normal BALD. In (10), the paper states that MJEnt = Ent + some additional term ("Posterior Uncertainty") = Epistemic Uncertainty + Aleatoric Uncertainty + Posterior Uncertainty, and that it thus decomposes three different uncertainties. This is wrong: it actually confounds these three terms (as it takes a sum/is a function of them)!

    The citation right before (4) and (5) is actually wrong. Gal et al, 2017, do not motivate the statement that follows.

3. Section 3.3 is not novel and only restates what is known from Houlsby et al, 2011, Gal et al, 2017, and Kirsch et al, 2019: entropy acquisition is over-confident compared to BALD because it adds in aleatoric (irreducible) uncertainty (Ent =  Epistemic Uncertainty (BALD) + Aleatoric Uncertainty), which will not be learnt away.

4. BalEnt falls out of nowhere as a ratio and is not actually motivated.

---

> ### Author Response · Authors · 2022-08-02
> **(continued from the previous comments)**
>
>
> - No justification of ratio term in BalEnt[x]
>   - **The reviewer's perception is wrong.**  As the reviewer *Swnb* agreed, Theorem 4.1 justifies the ratio term of BalEnt[x].
>
> - Relationship with BABA
>    - BABA[x] is a rescaled term of BalEntAcq[x]. i.e., $\frac{\text{BALD}[x]}{H(Y)+\log 2}\text{BalEntAcq}[x]=\text{BABA}[x]$ when $\text{MJEnt}[x]\geq 0$. The criticism of BABA[x] has been a lack of motivation in the ratio. However, BalEntAcq[x] is supported by Theorem 4.1.
>
> - $P4 = \text{BalEnt}[x]^{-1}$.
>    - We added P4 in Appendix A.16. P2 and P4 show almost the same performance because negative BalEnt[x] points have not been acquired in our experiments. However, it does not make sense once P4 reaches the negative sign. As P1 result suggests, adding a high magnitude negative BalEnt[x] point (which will be prioritized when P4 reaches the negative sign) should deteriorate the performance of active learning. So it's not a consistent way to prioritize points.
>
>
> [1] Gal, Yarin, and Zoubin Ghahramani. "Dropout as a bayesian approximation: Representing model uncertainty in deep learning." international conference on machine learning. PMLR, 2016.
>
> [2] MacKay, David JC. "Choice of basis for Laplace approximation." Machine learning 33.1 (1998): 77-86.
>
> [3] Cover, T.M. and Thomas, J.A., 2006. Elements of information theory 2nd edition.
>
> [4] Hein, Matthias, Maksym Andriushchenko, and Julian Bitterwolf. "Why relu networks yield high-confidence predictions far away from the training data and how to mitigate the problem." Proceedings of the IEEE/CVF Conference on Computer Vision and Pattern Recognition. 2019.

---

> ### Author Response · Authors · 2022-08-02
> **Thank you for raising many issues. However, we don't see any critical issues.**
>
> Thank you for raising many issues. We want to clarify and explain the details.
>
> - Regarding the conflict with "BABA: Beta Approximation for Bayesian Active Learning," here's our response.
>    - To the best of our knowledge, BABA has never been officially published. The reviewer might need to consider many other possibilities without any prejudice.
>       - We sent our private comment to Senior AC about addressing this issue.
>    - We also note that Theorem 3.1 provides more insight than the DirichletBALD[x] result of "Analytic Mutual Information in Bayesian Neural Networks." i.e., BetaMarginalBALD[x] tells us that BALD[x] is a function of marginals. Thinking about the dependency in Dirichlet distribution between marginals, it is not trivial to conclude this insight - BALD[x] is a function of marginals. This implies that the core information is embedded in the marginal distribution. Therefore the choice of MJEnt[x] (*marginalized* joint entropy) could be the key term in BalEntAcq[x].
>
> - Beta Approximation
>   - **We already explained the justification of the Beta approximation in Section 3.1.** Here's the short outline of the approximation. We are happy to answer more details if you are not convinced.
>       - We consider BNN as a proxy of the Gaussian process [1].
>       - Taking softmax would produce Dirichlet distribution [2].
>       - Each marginal of Dirichlet follows the Beta distribution.
>
>   - Therefore, our Beta approximation is well-supported by the theory. **One can always criticize that any parametric family might not fit perfectly.**  For example, people may criticize the linear regression model because data distribution is not linear. However, linear model tells us how those variables are linearly related. This is important because we can *understand* how different variables are linearly correlated. It's also crucial when we analyze the noisy dataset through linear models. Similarly, the Beta distribution family $\text{Beta}(\alpha,\beta)$ is one of the most richest and flexible parametric families to generate random values on $[0,1]$. This parametric modeling helps humans to explain/understand the behavior of the BNN. This way, we can derive and understand important insights explained in Section 3.4 and Appendix A.7-A.10 through closed forms.
>
> - Decomposition in MJEnt[x]
>   - After applying Jensen's inequality, **MJEnt[x] term is a valid decomposition by leveraging point process entropy on each marginal.** Please check the equation (8) in Appendix A.1. There's nothing wrong with our statement. *Decomposition does not require independency between terms.*
>    - MJEnt[x] involves both differential entropy (in Posterior Uncertainty) and Shannon entropy (in Epistemic+Aleatoric Uncertainties). With the current textbook information theory [3], there's no proper way to interpret the MJEnt[x] term involving differential entropy and Shannon entropy. Nevertheless, we have successfully proved Theorem 4.1. Theorem 4.1 provides the most profound insight into understanding the MJEnt[x] term. **Theorem 4.1 is a highly non-trivial statement.**
>
> - Section 3.3
>    - We understand similar observations with specified papers. However, **the nuances of "over-confident" and "over-fitting" are entirely different.** For example, there is a paper "Why ReLU networks yield high-confidence predictions far away from the training data and how to mitigate the problem" [4]. We can't say, "Why ReLU networks yield **over-fitted** predictions far away from the training data and how to mitigate the problem." In other words, "over-confident" is meant for the test data at the inference stage. And "over-fitting" is meant for the specific training data point at the training stage.

---

### Official Review · Reviewer_Swnb · 2022-07-18

**Rating:** 6
**Confidence:** 2
**Soundness:** 3 good
**Presentation:** 3 good
**Contribution:** 3 good

**Summary:**

By assuming a beta distribution on the softmax probabilities of a Bayesian classification network, the authors derive the marginalized joint entropy of a datapoint x and use it in a new acquisition function (balanced entropy learning acquisition) for active learning.

**Questions:**

Please see above

**Limitations:**

The authors didn't seem to discuss the limitations of the work although they claimed to have done that in the checklist (my apologies if I missed it somewhere).

**Strengths And Weaknesses:**

**Strength**
- The paper is well organized. The authors explained their assumptions carefully and also the method.
- Theorem 4.1 motivates the reciprocal of BalEnt well. Note that I haven't checked the proof in detail.

**Weakness and Question**
- I wonder how the assumption about the beta distribution of the softmax probabilities holds in practice. It would be beneficial to test this, for the dropout network, and for other types of Bayesian variational approximation methods.

- The authors didn't seem to discuss the limitations of the work clearly.

---

> ### Author Response · Authors · 2022-08-02
> **Thank you so much for validating our work.**
>
> Thank you very much for validating our work. We would be happy to answer any further details if you have any questions or concerns to increase your confidence.
>
> - Benefits of Beta approximation
>   - For empirical validation of Beta approximation, we already added an extensive numerical validation in Appendix A.8. It would be helpful to check this.
>   - We may understand the Beta approximation **as a calibration step of the predictive output.**
>     - By doing so, we have a much more detailed output distribution resolution (=density function) **with very few parameters**. So we can derive more tangible insights about uncertainties.
>     - For example, given $\text{Beta}(\alpha,\beta)$, the epistemic uncertainty is maximized as $\alpha,\beta\to 0$. Instead, the aleatoric uncertainty is maximized as $\alpha,\beta\to +\infty$.
>   - If we take a non-parametric approach such as ELR (please refer to the comments to the reviewer *fhr7*), it could give more flexibility. But typically, we end up with high computational cost problems.
>       - Instead, our Beta approximation facilitates a closed-form formula such as MJEnt[x]. So we were able to design a linearly-scalable acquisition measure.
>
> - Lack of Limitation
>    - We added the Limitation section that our experiments had been limited to dropout-based BNNs.
>
> Thank you very much again.

---

### Author Response · Authors · 2022-08-02
**Thank you very much for reviewing our paper.**

Thank you very much for reviewing our paper. After receiving all reviews, we realized that our work has been highly under-valued in the following aspects:
 - Most reviewers didn't validate our mathematical contributions properly. Our paper leverages a unique mathematical/or information-theoretic tool, a.k.a. **point process entropy**, to quantify the marginalized joint information between the model and the output.
    - **Point process entropy is neither Shannon entropy nor differential entropy. It is a new type of entropy in AI/ML community.**
    - We should NOT expect that the point process entropy is always similar to Shannon entropy or differential entropy. The behavior of the point process entropy is sometimes completely different from Shannon entropy (for discrete domain) and differential entropy (for continuous domain).
    - Our understanding of the point process entropy is still limited. For example, as shown in the form of *MJEnt[x]*, the derived form involves both the differential entropy and Shannon entropy. There has been no clear interpretation of the quantity which contains both differential entropy and Shannon entropy in the literature. **Our Theorem 4.1 connects two different entropies and provides a deep insight into the differential entropy and Shannon entropy simultaneously.**
    - Nevertheless, Beta approximation combined with point process entropy facilitates the theoretical study of Bayesian neural networks. That's how we could develop our own active learning method.
     - Again, **Theorem 4.1 is our main theorem** to justify the BalEntAcq[x] form as a ratio between MJEnt[x] and Shifted entropy. **We emphasize that this theorem is highly non-trivial.** We spent a significant amount of time developing Theorem 4.1 and ensuring the proof's correctness.

- To illustrate a better insight into our proposed method, we added **Appendix A.13**. Our proposed method BalEntAcq[x] is strongly aligned with active learning with abstention [1,2], which **guarantees to save labels exponentially for binary classification problems.** It has been well-known that active learning cannot perform better than passive learning (random selection) in general [3]. However, we can still achieve exponential savings on the labels by leveraging active learning with abstention. Therefore, these lines of work theoretically validate the benefits of active learning algorithms in practice. Our proposed method is well aligned with this theoretically guaranteed active learning algorithm (in terms of exponential savings).

So we revised and uploaded our main article and the supplementary material by addressing the majority of raised issues. We used the *red* color wherever we revised. *In the supplementary material, we also included our active learning source code under "active_learning_src".*
**We hope this rebuttal session helps you understand our core contributions properly, and we would be very grateful if you could re-evaluate our works. Thank you very much.**

[1] Zhu, Yinglun, and Robert Nowak. "Efficient Active Learning with Abstention." arXiv preprint arXiv:2204.00043 (2022).

[2] Puchkin, Nikita, and Nikita Zhivotovskiy. "Exponential savings in agnostic active learning through abstention." Conference on Learning Theory. PMLR, 2021.

[3] Castro, Rui M., and Robert D. Nowak. "Minimax bounds for active learning." IEEE Transactions on Information Theory 54.5 (2008): 2339-2353.

---

> ### Author Response · Authors · 2022-08-09
> **Thank you again for taking the time.**
>
> Thank you very much again for taking the time. We tried our best to address most of the concerns as much as we could. We hope that our rebuttal mitigates your concerns accordingly.

---

### Meta-Review · Area_Chair_iaj9 · 2022-08-26

**Recommendation:** Reject
**Confidence:** Certain

**Metareview:**

The majority of reviewers found this paper to be confusing in its presentation, lacking novelty (e.g. Section 3), and not well motivated (e.g. BalEntAcq), with 3 out of 4 recommending rejection.  I find that the paper particularly falters in its explanation of the point process entropy and derivation of the ultimate acquisition function.  As the author-review discussion makes clear, the deviation from Shannon / differential entropy to point process entropy is at the core of the paper, but this is lost in the current draft.  Due to this and moderate-to-minor issues such as the validity of the Beta approximation, the use of only MC dropout, and relationship to ELR schemes, I recommend rejection at this time.

**Award:**

No

---

### Decision · Program_Chairs · 2022-09-14

Reject